

# Prediction method of particle size qualification rate of high-pressure roller mill based on end-edge-cloud synergy

Hairong Guo[1], MingYin Yan[1], Jing Zhao[1] and Lanhao Wang[2]

[1] School of Mechanical Engineering, Shenyang University of Technology, Shenyang, Liaoning, China
[2] State Key Laboratory of Coking Coal Resources Green Exploitation, China University of Mining and Technology, Xuzhou, Jiangsu, China

## ABSTRACT

The pass rate of granules is an essential indicator during the high-pressure grinding process, as it accurately reflects the processing quality. Currently, the pass rate of granules is detected primarily based on manual experience judgments or offline inspections. Hence, this article presents a methodology for predicting the pass rate of granularity via an optimized support vector regression approach improved through genetic algorithms. Initially, a time-delay analysis method based on the particle swarm optimization algorithm is applied to mitigate the effects of time delays between the granularity pass rate and other data, thus aligning the dataset on a temporal scale. Subsequently, the feature data were selected using the maximum information coefficient analysis technique, which identified the most significant variables for inclusion in the training and testing sets of the predictive model. Predictions are then made using a support vector machine model that has been enhanced via genetic algorithm optimization. Furthermore, an online prediction model has been established, enabling real-time forecasting of the granularity pass rate and online model updates through root mean square propagation gradient descent optimization algorithm. This method leverages end-edge-cloud collaboration to provide a smart detection mechanism for the throughput rate of particles in high-pressure grinding mills. Experimental results demonstrate that, compared to traditional time-delay analysis, the improved time-delay analysis method proposed in this study is more effective and accurate. Simultaneously, the $\varepsilon$-GASVR granularity pass-rate prediction model proposed in this article achieved an $R^2$ of 0.89.

# INTRODUCTION

Given the current global economic development and the objective condition of non-renewable mineral resources, the efficiency and quality of the mineral processing industry have become essential factors in enterprise competition. China is rich in mineral resources but faces environmental and sustainable development challenges in mining and processing. Recently, many large and medium-sized metal mines in China have upgraded their coarse,

Corresponding author
Lanhao Wang,
wanglanhao888@163.com

medium, and fine crushing equipment, adopting the concept of more crushing and less grinding (*Riliang, Guangzhi & Yifei, 2011*). Thus, high-pressure grinding rolls (HPGR) have been increasingly applied in areas such as ore and slag. However, operating high-pressure roller mills and detecting particle size qualification rates is still challenging.

Traditional control methods for HPGR mainly rely on manual operation and offline inspection, which have high lag, low accuracy, and low efficiency. Especially for particle size qualification detection and control, traditional methods must meet the requirements of real-time, accurate, and efficient operations.

With the rapid development of information technology and industrial automation, technologies like cloud computing, the Internet of Things (IoT), and artificial intelligence (AI) are gradually penetrating various aspects of industrial production. End-edge-cloud collaborative technology has provided a new solution for particle size qualification detection of HPGR (*Yang, Liang & Ji, 2021*; *Zhou et al., 2021*). Specifically, the cloud-based intelligent control system utilizes algorithms and AI models to accurately predict the mill's working status and product quality through data analysis and learning capabilities. It then sends suggestions for adjusting parameters and control strategies to the equipment side. This two-way data communication and decision-making mechanism enables the detection system to achieve real-time and intelligent adjustments, improving particle size qualification rates while reducing waste and energy consumption in the production process.

This article presents a method that detects the particle size qualification rate of HPGR using end-edge-cloud collaboration. In addition, it discusses the crucial steps in cloud-based data analysis and model building and how these models are employed for real-time detection on the equipment side. Furthermore, the proposed method deploys an offline support vector machine model on the cloud side to predict the particle size qualification rate. In contrast, the edge side focuses on online data preprocessing and an online support vector machine prediction model for the particle size qualification rate. Additionally, the equipment side deploys a data acquisition and transmission module for the HPGR processing process. Furthermore, the article explores the advantages and challenges of this method and discusses its application prospects in high-pressure grinding rolls.

## LITERATURE REVIEW

In recent years, many scholars have adopted computer vision-based algorithms to detect ore particle size.many scholars have used visual algorithms to detect ore particle size. *Li et al. (2024)* proposed a new method to generate a synthetic dataset of particles and automatically annotate them using 'copy and paste' technology. This method creates a large and diverse particle dataset without the need for manual annotation. At the same time, the instance segmentation model Mask Region-based Convolutional Neural Network (Mask R-CNN) was improved, and a model specifically for particle instance segmentation was developed. The results were in good agreement with manual screening results, with errors within 7% for different types of particles.

*Cardoso et al. (2023)* proposed a machine vision concept based on edge artificial intelligence architecture and deep convolutional neural algorithm to achieve real-time

analysis of particle size as an alternative to offline laboratory processes. *Cheng et al. (2024)* designed a coal particle size analysis model based on a dual-layer routing attention mechanism (BRA). In order to reduce the problems of missed segmentation and over-segmentation in the coal particle segmentation process, feature information extraction is performed on the segmented coal particles, considering that the coal particle size in the coal particle dataset used in the experimental analysis is equivalent to the cell size. *Zhang, Feng & Zhang (2022)* proposed an improved watershed algorithm for online automated detection of ore particle size, and the cumulative error is within 3% compared with the results of manual screening. *Wang, Lian & Di (2021)* proposed a digital screening and detection method for the particle size of dam materials based on a combination of deep learning model and neighborhood component feature algorithm. This method can quickly detect the particle size distribution of the material heap by taking pictures of the material heap image. The results show that the feature extraction and particle size detection accuracy have been improved.

Vision algorithms, which are employed as granularity detection technologies rooted in image processing and computer vision, exhibit numerous advantages.

Despite their substantial benefits, vision algorithms exhibit certain limitations when applied to practical engineering tasks. Firstly, their effectiveness can be compromised by environmental conditions, such as variable lighting and the presence of dust, which may significantly reduce image quality and, as a result, impair detection accuracy. Secondly, the inherently complex nature of these algorithms often requires significant computational resources, which could lead to processing lags that are ill-suited for real-time monitoring applications. Additionally, the acuity of vision algorithms is limited by the resolution of the images they process. This impediment can make it challenging to detect very small particles, particularly those that have been altered by industrial processes such as lamination in a high-pressure roller mill, where finer details may be beyond the detection capability of the algorithms alone and may necessitate the use of additional methods like sensor data analysis for accurate identification.

At present, the algorithm model constructed using analog quantity to make forecasts has matured. *Azizi, Rooki & Mollayi (2020)* investigated the application of three powerful kernel-based supervised learning algorithms to develop a global model of the wear rate of grinding media. It is distinguished that compared to the single kernel and ANN-based techniques, the use of multiple kernel support vector machines benefit from a higher degree of correctness and generalization ability for prediction of wear rate of grinding media. In the research of *Ke et al. (2021)*, a novel intelligent model was proposed to predict ground vibration intensity based on the hybridization of autoencoder neural networks (AutoencoderNN) and support vector machine regression (SVR), and it was named Autoencoder NN-SVR. The study of *Li et al. (2020)* is combined support vector machine and improved dragonfly algorithm to forecast short-term wind power for a hybrid prediction model. The adaptive learning factor and differential evolution strategy are introduced to improve the performance of traditional dragonfly algorithm. The improved dragonfly algorithm is used to choose the optimal parameters of support vector machine. In *Jannumahanthi & Murugesan (2020)*, a detailed study of diesel engine performance

using support vector regression and performance metrics such as brake thermal efficiency and accuracy are explored. Findings specify that support vector regression is an efficient technique for diesel engine performance that validates and compares the actual performance with high accuracy.

## MATERIALS & METHODS

### Overview of high-pressure roller grinding process

This article selects the process flow of the HPGR for a specific beneficiation on-site plant as follows: the ore enters the material field after being screened by a middle crushing screen. It then enters the buffer silo of the HPGR through a feeding belt and undergoes processing in the HPGR. The buffer silo serves as a transition and reserve to ensure the continuous and stable operation of the HPGR. The principle of the HPGR crushing process is laminating crushing, (*Bo et al., 2022*) and the ore is confined between the rolls, begins to aggregate, and is compressed with the rotation and pressure applied to the registrations. Due to the high tension between the rolls and the mutual collision and compression of the ore particles, the ore is entirely crushed to the required particle size (*Rashidi, Rajamani & Fuerstenau, 2017*; *Aydoğan, Ergün & Benzer, 2006*).

During the operation of the HPGR, the scattering crushing force on the rolls disperses the ore, improving the energy efficiency and fineness of the crushing machine. After processing, the ore is discharged from the outlet and transported to the buffer silo *via* a feeding belt. The buffer silo serves as a stable feeding device for the following process, ensuring the continuous operation of the production line. The ore then enters a differential screen and, after passing through a fine mesh, delivers the qualified ore to the following process *via* a belt, while the unqualified material is returned to the material field through another belt, forming a closed-loop grinding process.

Figure 1 illustrates the proposed process covering the entire closed-loop processing process from the feeding belt to the HPGR for processing and then to the screening of qualified materials and unqualified returns *via* a fine screen. The feeding amount is controlled by adjusting the opening of the material control plate and the displacement of the feeding belt throttle valve. This changes the processing quantity, material weight, and roll clearance to improve accuracy while ensuring processing efficiency.

### Current status of the particle size qualification rate detection process

Manual operation requires much time and effort but also needs to improve its response, making it challenging to capture the changes in particle size in a timely and accurate manner. In addition, manual control is known for its low efficiency, low accuracy, and high latency, which may result in untimely adjustments and pose a risk of malfunctions in the high-pressure roller mill. Given that there is a typical interconnection between different processes in the related industries, when an abnormality occurs in the high-pressure roller mill, it disrupts the production rhythm of this process. It interrupts the upstream feeding and downstream discharging systems. Hence, the entire production flow is affected. Therefore, fine processing and product output must be improved, leading to decreased

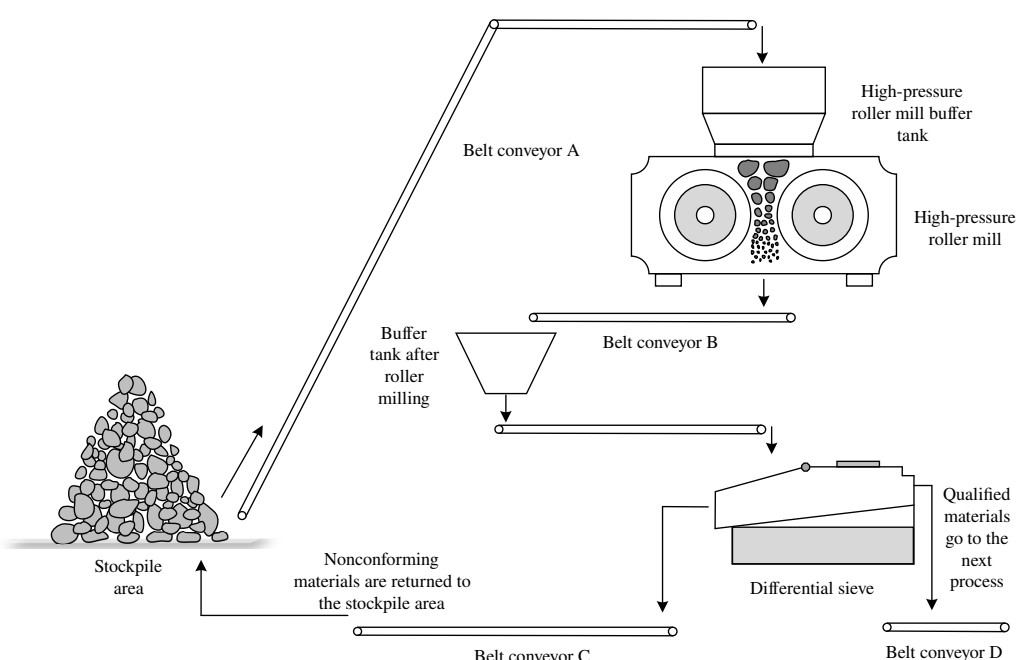

**Figure 1** Schematic diagram of high-pressure roller grinding process flow.

production and even production shutdown, resulting in significant economic and time losses.

## Detection strategy for the qualified rate of particle size in high-pressure roller mills

This article proposes a new method for predicting the granule qualification rate of high-pressure roller mills based on end-edge-cloud collaborative technology. The developed technique combines a genetic algorithm with support vector regression and updates the online model using the gradient descent algorithm based on RMSprop. In addition, the data acquisition process of the high-pressure roller mill processing is significant and measurable. Moreover, the edge system is responsible for data processing and the granule qualification rate prediction online model. The cloud system includes data servers and AI computing platforms, while the data collected involve the roller gap width $R(k)$, clamping pressure $L(k)$, roller frequency $F(k)$, bin weight $W(k)$, and processing volume $C(k)$. The input of the online prediction model produces the predicted result $P_p(k)$, which corresponds to the actual value of the granule qualification rate $P_p(k+k_z)$. Based on the self-correction mechanism, when the actual value and the predicted value do not meet the error requirements, the prediction accuracy of the online support vector machine model is improved using the self-correction mechanism. When the accuracy requirement is not met, the parameter calibration of the support vector machine model is performed online using the self-correction support vector machine model's hyperparameters to ensure the prediction accuracy of the granule qualification rate of the high-pressure roller mill. The

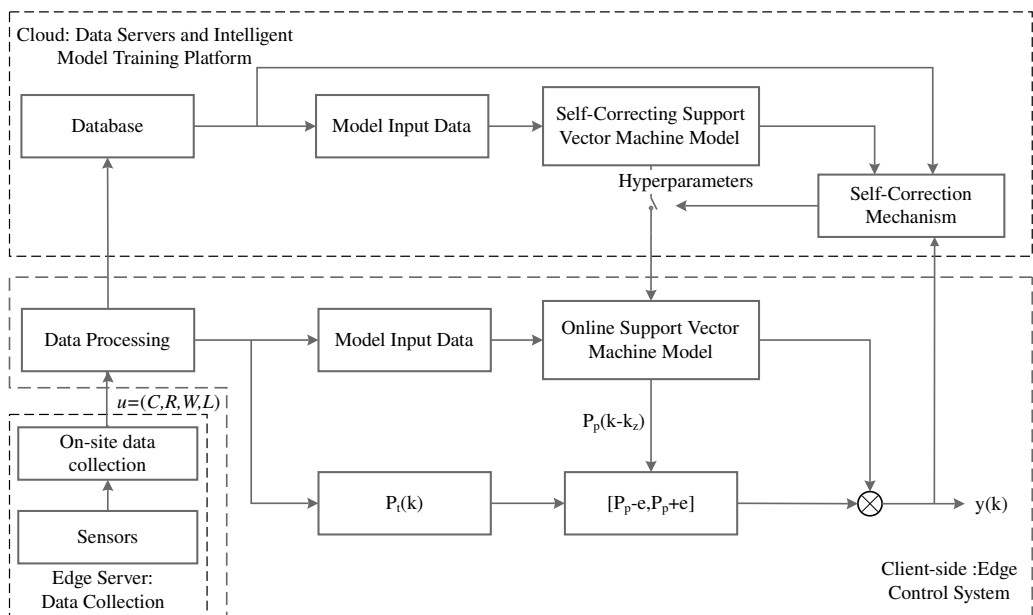

**Figure 2** Architecture for predicting the qualified rate of particle size in high-pressure roller mills based on edge-cloud collaboration.

predictive framework for the particle size qualification rate of high-pressure grinding rollers in end-edge-cloud collaboration is illustrated in Fig. 2.

## Time delay analysis based on particle swarm optimization algorithm

Due to the lag in the detection data of conveyor belt transportation during the high-pressure roll grinding process, the current granular qualification rate obtained at any given moment is processed under the working conditions before a time delay of t. Therefore, a time delay analysis is mandatory to align the other sensor data with the obtained granular qualification rate on a temporal scale. The traditional timing delay analysis methods primarily determine the delay time by calculating the correlations between input and output variables at different moments. However, due to the specificity of the high-pressure grinding roller (HPGR) system, traditional methods exhibit two shortcomings: firstly, in the system, a single measurement is not only associated with the particle size pass rate but is also affected by other variables, equipment, and manual control. Secondly, the delay time of correlated variables may change under different operational conditions, given that conditions vary across different working states. In response to these requirements, we introduce a time delay analysis method based on a particle swarm optimization algorithm.

Particle swarm optimization (PSO) is an evolutionary computing technique inspired by the social behavior of bird flocks (*Jain et al., 2022*), particularly their foraging patterns. Within PSO, solutions are depicted as "particles" that navigate the search space as their environment. Particles dynamically adjust their positions within the solution space, drawing from their own experiences and that of nearby particles. These adjustments are influenced by a particle's velocity, which is updated to reflect the best positions found by the individual

and the swarm as a whole. This capability enables the swarm to locate both global and local optima, thus enhancing the quality of delay analysis.

In addressing delay issues, PSO is employed to identify the most effective timing synchronization sequence for the network's nodes. The optimization targets are the delay vectors—each node's relative delay to a reference time source—and the particle swarm is utilized to traverse these vectors within a multidimensional search space. The PSO method excels in this context, adept at managing multimodal challenges and pinpointing delay vectors that mitigate network synchronization errors. The procedural steps to perform delay analysis *via* the Particle Swarm Optimization algorithm are detailed subsequently.

Let t be the time $X^t = [x_0^t, x_1^t, \ldots, x_i^t, \ldots, x_N^t]$, and there exist different true time delays between different process variables and $Q_t$. The delay vector is $T_t$. Assuming at time point t, the optimal delay vector is estimated from a large amount of historical data.

$$T_t' = [\tau_1, \tau_2 \cdots, \tau_i, \ldots, \tau_N] = [\delta_1 T, \delta_2 T, \ldots, \delta_i T, \ldots, \delta_N T] \tag{1}$$

where T is the sampling period, is the delay reference for $x_i$, and then reconstruct

$$X^t = [x_0^t, x_1^{t+\tau_1}, \ldots, x_i^{t+\tau_i}, \ldots, x_N^{t+\tau_N}] \tag{2}$$

Select $X_{x0}$ starting at time t for q sampling periods of T

$$\widetilde{x_0} = [x_0^t, x_0^{t+T}, x_0^{t+2T}, \ldots, x_0^{t+(q-1)T}]^T \tag{3}$$

$$\widetilde{x_i} = [x_i^{t+\delta_i T}, x_i^{t+(\delta_i-1)T}, x_i^{t+(\delta_i-2)T}, \ldots, x_i^{t+(\delta_i-(q-1))T}]^T \tag{4}$$

Q and t should not be too small and must be greater than the empirical value of the time interval between the moment the high-pressure roller mill is unloaded and the moment the material comes out. Based on this, a multi-variable correlation reconstruction matrix is constructed:

$$\widetilde{X} = [\widetilde{x_0}, \widetilde{x_1}, \ldots, \widetilde{x_N}]$$
$$= \begin{bmatrix} x_0^t & x_1^{t+\delta_1 T} & \cdots & x_N^{t+\delta_N T} \\ x_0^{t+T} & x_1^{t+(\delta_1-1)T} & \cdots & x_N^{t+(\delta_N-1)T} \\ \vdots & \vdots & \vdots & \vdots \\ x_0^{t+(q-1)T} & x_1^{t+(\delta_1-(q-1))T} & \cdots & x_N^{t+(\delta_N-(q-1))T} \end{bmatrix}_{q \times (N+1)} \tag{5}$$

$\Lambda_i = \widetilde{X}_0 - \widetilde{X}_i, i = 1, 2, \ldots, N, \Lambda$ is a q ×N Matrix, construct the gray correlation coefficient matrix H:

$$H_{ji} = \frac{\min\{\Lambda_{ji}\} + \alpha \max\{\Lambda_{ji}\}}{\Lambda_{ji} + \alpha \max\{\Lambda_{ji}\}} j = 0, 1, \ldots, q-1; i = 1, 2, \ldots, N \tag{6}$$

$\alpha$ is usually set to 0.5. Obtain the gray correlation degree:

$$O = \frac{1}{q} \sum_{i=1}^{N} \sum_{j=0}^{q-1} H_{ji} \tag{7}$$

The larger the O value, the closer the estimated delay vector T' is to T to some extent.

Then, the optimal T' is obtained based on the particle swarm optimization algorithm. Given N process variables that need to estimate the time delay relative to the reference variable x0, the particle swarm needs to search for the optimal delay base vector in an N-dimensional space, and the fitness function for evaluation be the gray correlation degree O. Let the i-th particle position vector in the N-dimensional space be:

$$x^i(k) = [x_1^i, x_2^i, \ldots, x_n^i] \tag{8}$$

The velocity vector of the i-th particle is

$$v^i(k) = [v_1^i, v_2^i, \ldots, v_n^i] \tag{9}$$

The velocity update formula for the i-th particle is:

$$v_j^i(k+1) = \omega(k)v_j^i(k) + c_1 r_1(p_j^i(k) - x_j^i(k)) + c_2 r_2(p_j^g(k) - x_j^i(k)) \tag{10}$$

The position update formula for the i-th particle is *Cui (2022)*, *Jia et al. (2023)*:

$$x_j i(k+1) = x_j i(k) + v_j i(k+1) \tag{11}$$

where k is the current iteration number; $r_1$, $r_2$ are uniformly distributed random numbers in the range [0, 1]; C1, C2 are learning factors; $p_j^i(k)$ is the individual best position of the i-th particle before k iterations; $p_j^g(k)$ is the global best position of the i-th particle swarm before k iterations;

$$\omega = \omega_{max} - (\omega_{max} - \omega_{min})k/G \tag{12}$$

$\omega$ is the inertia weight, $\omega$max is the maximum inertia weight, $\omega$min is the minimum inertia weight, G is the maximum number of iterations. Let the end condition for the particle swarm iteration be that the difference between the fitness value of the best solution after the previous iteration and the current iteration is less than e. Obtain the particle position vector corresponding to the optimal fitness value, which represents the true time delay vector T between multiple variables. By doing so, the consistency of granularity qualified rate data with other sensor data can be maintained.

### $\varepsilon$-support vector regression (SVR) prediction algorithm based on genetic algorithms

(1) Support vector regression

Support vector regression (SVR) is a regression method based on Support Vector Machine. Unlike traditional regression methods, SVR seeks a boundary that minimizes the error between the predicted results and the actual values of the training samples while considering the principle of minimum structural risk (*Ping et al., 2016*; *Bao, 2022*)). $\varepsilon$-SVR defines a boundary by introducing a tolerance range $\varepsilon$, which minimizes the error between the predicted results and the actual values of the training samples. However, $\varepsilon$-SVR still has parameter sensitivity, computational complexity, and consistency issues. Selecting the algorithm parameters directly affects the model's prediction accuracy, and combining grid search and cross-validation for a support vector machine parameter optimization algorithm

is time-consuming. Furthermore, this strategy may not yield the globally optimal solution. The genetic algorithm (GA) is adopted as a robust and intelligent identification method that operates directly on the optimization object and presents an excellent global optimization ability and parallelism.

In the following experiment, the $\varepsilon$-GASVR, which is based on a genetic algorithm, partially compensates for the shortcomings of $\varepsilon$-SVR. The improved genetic algorithm is used for parameter optimization, demonstrating significant advantages over the grid search algorithm in strengthening model prediction accuracy.

For the nonlinear function regression problem, let there be n data samples $\{x_i, y_i\}$, where $x_i$ isthe N-dimensional sample input, $x_i \in RN$, $y_i$ is the sample output, $y_i \in R$. Through nonlinear transformation, the N-dimensional input data is mapped to the high-dimensional feature space F, and the optimal linear regression function is constructed in this space (*Hongyang, Peng & Yunzhe, 2023*):

$$f(x) = \omega^T \Phi(x) + b \tag{13}$$

where T represents transpose; $\omega$ and b are the normal vector and offset of the regression function, respectively.

$\varepsilon$-SVR introduces a tolerance range $\varepsilon$ to define a boundary, allowing the predicted results to fluctuate within the scope of $\varepsilon$ and be considered valid. Therefore, the goal of the model is to minimize the following loss function:

$$\min \frac{1}{2}\|\omega\|^2 + C \sum_{i=1}^{L}(\xi_i + \xi_i^*) \, s.t. \begin{cases} \omega^T x_i + b - y_i \le \varepsilon + \xi_i \\ y_i - (\omega^T x_i + b) \le \varepsilon + \xi_i^* \\ \xi_i, \xi_i^* > 0, i = 1, 2, \cdots, n \end{cases} \tag{14}$$

where $\|\omega\|$ is the norm of the weight vector $\omega$, C is a hyperparameter used to control the complexity of the model, and $\xi$ and $\xi^*$ are slack variables used to handle noise and outliers. $\xi$ represents the error of samples within the $\varepsilon$ band, and $\xi^*$ represents the error of samples outside the $\varepsilon$ band.

After introducing the Lagrange function:

$$L(\omega, b, \xi_i, \xi_i^*) = \frac{1}{2}\omega^T\omega + C\sum_{i=1}^{n}(\xi_i + \xi_i^*) - \sum_{i=1}^{n}\alpha_i(\omega^T\Phi(x_i) + b - y_i + \varepsilon + \xi_i) -$$
$$\sum_{i=1}^{n}\alpha_i^*(y_i - \omega^T\Phi(x_i) - b + \varepsilon + \xi_i^*) - \sum_{i=1}^{n}(v_i\xi_i + v_i^*\xi_i^*) \tag{15}$$

In the formula, $\alpha_i$ and $\alpha_i^*$ are Lagrange multipliers. Where $\alpha_i, \alpha_i^* \ge 0$, $\gamma_i, \gamma_i^* \ge 0$, 1, ..., n. When the partial derivatives of each variable are 0.

$$\begin{cases} \omega = \sum_{i=1}^{n}(\alpha_i + \alpha_i^*)\Phi(x_i) \\ \sum_{i=1}^{n}(\alpha_i + \alpha_i^*) = 0 \\ v_i = C - \alpha_i \\ v_i^* = C - \alpha_i^* \end{cases} \tag{16}$$

According to the duality principle and kernel function technique:

$$\min \frac{1}{2}\sum_{i=1}^{n}\sum_{j=1}^{n}(\alpha_i-\alpha_i^*)(\alpha_j-\alpha_j^*)k(x_i,x_j)+\varepsilon\sum_{j=1}^{n}(\alpha_i+\alpha_i^*)-\sum_{j=1}^{n}(\alpha_i-\alpha_i^*)s.t.$$

$$\begin{cases} \sum_{j=1}^{n}(\alpha_i-\alpha_i^*)=0 \\ 0\leq\alpha_i,\alpha_i^*\leq C, i=1,2,\cdots,n \end{cases} \quad (17)$$

A key issue in nonlinear regression is the choice of the kernel function, which initially applies a nonlinear mapping to project the data into a high-dimensional feature space and then performs regression within that space (*Widodo & Yang, 2007*). The selection of the kernel function directly influences the accuracy of the predictive model. Common types of kernel functions include the linear kernel, the polynomial kernel, and the Gaussian radial basis function (RBF) kernel. The Gaussian RBF kernel is expressed as follows:

$$K(x_i,x_j)=\tanh(\gamma(x_i\cdot x_j)+r) \quad (18)$$

Here, $\gamma$, d, and r are kernel parameters, and the regression function can be obtained as:

$$f(x)=\sum_{i\in SV}(\alpha_i-\alpha_i^*)k(x_i,x)+b \quad (19)$$

$$\begin{cases} b=y_i-\omega^T x_i-\varepsilon, \alpha_i\in(0,C) \\ b=y_i-\varepsilon^T x_i+\varepsilon, \alpha_i^*\in(0,C) \end{cases} \quad (20)$$

where SV represents the support vector set, b is the bias obtained by solving for support vectors $(x_i, y_i)$ and allowing for a deviation $\varepsilon$. (2) Improved Genetic Algorithm

Genetic algorithms have some issues during the optimization process. Especially in the early stages of algorithm iteration, due to the scarcity of excellent individuals in the initial population, they are prone to be repeatedly selected as parent individuals, resulting in limited changes in offspring individuals, causing the algorithm to get stuck in a local optimum. Individual differences are relatively small in the later stages of iteration, and the algorithm may exhibit slow convergence or even non-convergence.

Spurred by these concerns, this study adopts a dynamic setting approach to adjust the genetic algorithm's crossover and mutation probability. Precisely, as the algorithm iterates, the crossover and mutation probability are adjusted accordingly. In the early stages of the algorithm's iteration, a higher crossover probability and a lower mutation probability are adopted to increase individuals' diversity and exploration ability. This encourages individuals in the population to explore the search space better and find better solutions. As the iteration progresses, the crossover probability gradually decreases while the mutation probability gradually increases. This introduces more randomness and diversity, allowing individuals to escape local optima and further optimize the quality of the solutions. By dynamically adjusting the crossover probability and mutation probability, the genetic algorithm can better balance the trade-off between exploration and exploitation, thereby improving the global search capability of the algorithm (*Tabassum & Mathew, 2014*).

In genetic algorithms, it is essential to adjust the crossover probability to maintain a balance between exploration and exploitation. To achieve this, we initially define a maximum crossover probability. As the algorithm iterates, we modulate this probability using a function of the standard deviation of fitness values within the population, which ensures a higher crossover probability during the early evolutionary stages to promote diversity, and a decreased probability in the later stages to facilitate convergence on an optimal solution. The precise relationship is governed by the following formula, which calculates the adaptive crossover probability:

$$P_c = P_{c\max} - \sqrt{\frac{\sum_{i=1}^{N}(f_i - f_{avg})^2}{N}} \cdot P_{ca} \tag{21}$$

In the described formula, each symbol is defined as follows:

$P_{cmax}$ represents the maximum crossover probability, serving as a ceiling for how high the probability can be adjusted.

$f_i$ denotes the fitness value of the $i^{th}$ individual within the population.

$f_{avg}$ is the average fitness value across all individuals in the population, acting as a benchmark for comparison.

N symbolizes the size of the population, a key factor in determining population diversity.

$P_{ca}$ is the adaptive parameter for crossover probability, adjusting the rate based on evolutionary progress and diversity needs.

In genetic algorithms, the mutation probability is crucial for introducing variability into the population. We begin by establishing a minimum mutation probability. Contrary to crossover probability adjustment, we progressively increase the mutation probability as the algorithm iterates, which is contrary to standard deviation trends of the population's fitness values. This approach encourages a more explorative search in the initial stages and a more exploitative search in the final stages, helping to prevent premature convergence on local optima. The adaptive mutation probability is detailed by the following specific formula:

$$P_m = P_{m\min} - \sqrt{\frac{\sum_{i=1}^{N}(f_i - f_{avg})^2}{N}} \cdot P_{ma} \tag{22}$$

In the formula, $P_{mmin}$ denotes the minimum mutation probability, and $P_{ma}$ signifies the mutation probability adjustment parameter.

## Online self-correction model for particle size qualified rate prediction

To facilitate real-time online prediction of the acceptance rate for particle size produced by the high-pressure roll grinding machine in a mineral processing plant, we employ the offline-established epsilon-Greedy Adaptive Support Vector Regression ($\varepsilon$-GASVR) model as the basis for online application. Once the $\varepsilon$-GASVR model is trained offline, it is deployed in the online setting along with its model parameters and configuration.

The relationship between current and historic particle size acceptance rates is governed by an observed time delay, such that $P_t(t_0)$, the present rate, corresponds to the past acceptance rate $P_p(t_0 - t_z)$. For incoming data, we conduct a fitness evaluation to ascertain

compatibility with the extant model; specifically, we verify whether $P_t(t_0)$ falls within the defined range $[P_p(t_0 - t_z) - \varepsilon, P_p(t_0 - t_z) + \varepsilon]$. Should $P_t(t_0)$ reside outside this range, we proceed to update the model parameters online, utilizing both $P_t(t_0)$ and feature data from $t_0 - t_z$.

Post-training, the prediction model acquires continual sensor data to make forecasts. To accommodate new information and adapt to evolving conditions, the online model is endowed with self-learning capabilities to refine its predictive precision autonomously.

The unpredictable and variable nature of mineral processing conditions necessitates that the input–output data for the online model be managed within a dynamic and nebulous information space. When notable changes occur, maintaining prediction accuracy becomes challenging. To address this, we introduce an online self-correction support vector machine model that utilizes the hyperparameters preset from the offline model. These hyperparameters are fine-tuned in real-time using the Root Mean Square Propagation (RMSprop) gradient descent optimization algorithm, which leverages exponentially weighted moving averages to dynamically adjust the learning rate and parameter step size. The RMSprop updates are formulated as follows:

$$S = \rho S + (1 - \rho)g^2 \tag{23}$$

In the described formula, each symbol is defined as follows:

S denotes the cumulative variable.

$\rho$ represents the decay rate, ranging between 0 and 1, which determines the influence of historical gradients on the current cumulative variable.

g signifies the gradient obtained from the recent sample.

Following this, the update step size for each parameter, denoted by $\Delta w$, is computed based on the cumulative variable S, thus affecting the magnitude of adjustments made to the parameters during the update. The specific formula for calculating $\Delta w$ is as follows:

$$\Delta w = \frac{-\eta g}{\sqrt{S} + \varepsilon} \tag{24}$$

In the described formula, each symbol is defined as follows:

$\Delta w$ stands for the parameter's update step size.

$\eta$ is the learning rate, and $\varepsilon$ is a small constant introduced to avoid a division by zero error. Subsequently, the parameter update step size $\Delta w$ is applied to adjust each parameter's value, steering the model closer to the optimal solution. The equation that delineates the process for updating the model parameters can be articulated as:

$$w' = w + \frac{-\eta g}{\sqrt{S} + \varepsilon} \tag{25}$$

$w$ represents the model parameter vector, and the initial model parameter vector consists of hyper-parameters of the offline support vector machine model. During the iteration process of the algorithm, the model performance is optimized by continuously updating the parameter vector w.

The self-correction mechanism adopts three indicators: prediction error, true positive rate (TPR), and true negative rate (TNR) of the online support vector machine prediction

**Table 1 Calculation methods for TP, FP, TN, and FN.**

| Condition | $P_p(k) - P_p(k-1) \geq 0$ | $P_p(k) - P_p(k-1) \geq 0$ |
|---|---|---|
| $P_t(k_z) - P_t(k_z - 1) \geq 0$ $P_t(k_z) - P_t(k_z - 1) \geq 0$ | $TP(k) = 1$ $FP(k) = 1$ | $FN(k) = 1$ $TN(k) = 1$ |

model (*Gao & Chai, 2023*).

$$\Delta P(k) = \left| P_p(k) - P_t(k_z) \right| \tag{26}$$

$$TPR(k) = \frac{\sum_{i=1}^{k} TP(i)}{\sum_{i=1}^{k} TP(i) + \sum_{i=1}^{k} FP(i)} \tag{27}$$

$$TNR(k) = \frac{\sum_{i=1}^{k} TN(i)}{\sum_{i=1}^{k} TN(i) + \sum_{i=1}^{k} FN(i)} \tag{28}$$

In the equation, the calculation methods for TP, FP, TN, and FN are shown in the Table 1.

When the prediction error $\Delta P(k)$ of the online support vector machine is either greater than or equal to $\varepsilon$, or less than $\varepsilon$ and the forecast accuracy of both the upward and downward trends of the online support vector machine model is less than or equal to 95%, the gradient descent method based on RMSprop is adopted to update the hyperparameters of the $\varepsilon$-GASVR model.

## Data modeling experiment

(1) Feature selection

In the domain of various feature selection methodologies, this study has employed mutual information-based feature selection (MIFS), also known as MIC correlation analysis. Compared to alternative feature selection approaches, MIC displays a heightened resilience to noise and outliers, and its performance remains less influenced by extreme values. This robustness ensures stable efficacy across diverse datasets and various sample sizes. Unlike methods that are highly model-dependent, MIC is model-independent, making it versatile enough to integrate with different machine learning algorithms, unconstrained by any specific modeling assumption. Moreover, MIC excels by not only discerning linear associations but also by capturing non-linear relationships between variables. This attribute renders it superior to traditional correlation measures when it comes to uncovering complex interdependencies among features. The calculation formula for MIC is as follows:

$$MIC[x;p] = \max_{|X||P|} \frac{I[X;P]}{\log 2(\min(|X|,|P|))} \tag{29}$$

$X$ symbolizes the other variables and $P$ denotes the granularity pass rate

The study considers data from eight measurements gathered *via* sensors equipped on a high-pressure roller mill during its operation, excluding data concerning processing

**Table 2  Correlation analysis results of MIC.**

| Variable names | Related coefficient |
|---|---|
| Driven side seam | 0.761 |
| Locking pressure | 0.694 |
| Dynamic roller frequency | 0.632 |
| Hopper weight | 0.630 |
| Fixed roller current | 0.510 |
| Fixed roller frequency | 0.505 |
| Dynamic roller current | 0.463 |
| Non-driven side seam | 0.310 |

capacity and particle size qualification rate. These measurements encompass the motor current of the movable roll, motor current of the fixed roll, hopper weight, frequency of the movable roll, frequency of the fixed roll, clamping pressure, gap on the drive side roll, and gap on the non-drive side roll. Rankings of the correlation coefficients, as determined by our MIC correlation analysis, are presented in Table 2.

Consequently, the selected features for this experiment include roll gap width R(k), clamping pressure L(k), roll frequency F(k), hopper weight W(k), and processing capacity C(k).

(2) Normalization processing

The experiment utilizes data accrued from the equipment on-site at a designated beneficiation plant. The dataset comprises a total of 10,000 data sets, partitioned into a training set and a testing set at a ratio of 7:3. To mitigate the impact of differing magnitudes across variables and to allow the data-driven model to concentrate on the intrinsic pattern within the data, it is imperative to execute Z-score normalization on the dataset prior to training.

The Z-score normalization process adopted in this research involves initially calculating the mean and standard deviation for each feature.

$$\mu = [\mu_1, \mu_2, \mu_3, \mu_4, \mu_5] \tag{30}$$

As for them, they are respectively the mean of each column in the matrix. Calculate the standard deviation vector for each feature.

$$\sigma = [\sigma_1, \sigma_2, \sigma_3, \sigma_4, \sigma_5] \tag{31}$$

Among them, they are respectively the standard deviation of each column in the matrix. Normalize each element of the matrix.

$$z_{ij} = \frac{(x_{ij} - \mu_j)}{\sigma_j} \tag{32}$$

Let $Z_{ij}$ represent the normalized value of the element in the i-th row and j-th column within the matrix, where $x_j$ is the original element at the corresponding position in the data matrix. Here, $\mu_j$ is the mean and $\sigma_j$ is the standard deviation of the j-th column. This process enables independent Z-score normalization of each feature in the matrix,

standardizes the variable scales, and resolves the issue of dimensional discrepancies between different features. Notably, this is particularly useful for certain machine learning algorithms, especially those contingent on linear relationships and relative scaling.

(3) Median filtering

In the context of sensor data processing, median filtering stands as a prevalent technique, proficient at mitigating the effects of noise on data analysis and subsequent decision-making processes. This technique endeavors to replace the current data point with the median of the surrounding data sequence, effectively curtailing disturbances wrought by outliers and sporadic noise impulses.

The fundamental concept underlying median filtering is the employment of a 'window size' that traverses the data sequence awaiting filtration. This 'window' is a dynamic, sliding construct, the length of which is tailored to align with the demands of the specific application at hand. For every position of the window, median filtering ascertains the median value of all encompassed data points, and adopts this median as the representation for the current data point under scrutiny. The distinct advantage this method harbors, as opposed to alternative filtering techniques, lies in its resilience against sudden, isolated noise events and its fidelity in conserving the edge features of the original signal. These attributes collectively enhance the precision of subsequent algorithmic interpretations and decision-making procedures.

To elucidate with an example, consider a data sequence S subject to filtering: should the size of the window be k and the central position of this window be i, then the filtered data point can be denoted as:

$$S = [S_1, S_2, S_3, \ldots, S_n] \tag{33}$$

Given a window size of k and the center position i, the filtered data point can be represented as:

$$y_i = median(S_i - \frac{1}{2}(k-1), S_i - \frac{1}{2}(k-3), \ldots, S_i, \ldots, +\frac{1}{2}(k-3), S_i + \frac{1}{2}(k-1)) \tag{34}$$

where $y_i$ represents the filtered data point, the median function calculates the median of the data points in the window. This formula represents the basic calculation process of median filtering. By traversing each data point $S_i$ and applying the window for filtering, the filtered data sequence can be obtained, which is

$$y = [y_1, y_2, y_3, \ldots, y_n] \tag{35}$$

To assess the feasibility and superior performance claims of the proposed algorithm, three predictive models were compared: GA-MLP, $\varepsilon$-SVR, and $\varepsilon$-GASVR. For the $\varepsilon$-SVR model, chosen for its support vector machine regression capabilities, a radial basis function (RBF) was selected as the kernel. The optimally tuned hyperparameters included a tolerance interval set to 0.001, a regularization parameter C fixed at 15, a kernel function parameter $\gamma$ at 0.5, and a tolerance level at 0.01. The $\varepsilon$-GASVR model was parameterized with a genetic algorithm population size of 500, a crossover probability of 0.8, mutation probability of 0.05, and a termination criterion of 1,000 iterations. In the GA-MLP model, used for

multilayer perceptron regression, the genetic algorithm parameters mirrored those of $\varepsilon$-GASVR, with an architecture featuring a single hidden layer with 150 neurons, the incorporation of the ReLU function as the activation mechanism, and settings including an output neuron count of 1, a learning rate of 0.001, a batch size of 256, and a total of 2,000 training iterations. For comparative analysis, all three models were trained and tested using an identical dataset comprising 10,000 samples.

## RESULTS

This article employs the mean absolute error (MAE), root mean squared error (RMSE), and mean absolute percentage error (MAPE) as evaluation metrics for the models. These metrics are defined as follows:

$$MAE = \frac{1}{n}\sum_{i=1}^{n}(p_p(i) - p_t(i)) \tag{36}$$

$$RMSE = \sqrt{\frac{1}{n}\sum_{i=1}^{n}(p_p(i) - p_t(i))^2} \tag{37}$$

$$MAPE = \frac{1}{n}\sum_{i=1}^{n}\left|\frac{(p_p(i) - p_t(i))}{p_t(i)}\right| \tag{38}$$

The coefficient of determination ($R^2$) also measures the goodness-of-fit of linear regression predictions to actual data. A higher $R^2$ value (closer to 1) indicates a better fit and, thus, a better linear regression model.

$$R^2 = 1 - \frac{\sum_{i=1}^{n}(p_p(i) - p_t(i))}{\sum_{i=1}^{n}(p_t(i) - \overline{p_t(i)})} \tag{39}$$

Traditional time delay analysis methods primarily determine the delay duration by calculating the correlation between input and output variables at different times. However, due to the specific nature of high-pressure roller mill systems, traditional methods exhibit two main drawbacks: First, in the system, a single measurement is not only related to the granularity pass rate but is also influenced by other variables as well as equipment and manual control. Second, different operating conditions may lead to changes in the delay times of the correlated variables.

The experiment utilized a dataset comprising 10,000 samples, which included parameters such as driven Side Seam, locking Pressure, dynamicRoller Frequency, hopper Weight, fixed Roller Current, fixed Roller Frequency, dynamic Roller Current, non-driven Side Seam in correlation with the granularity pass rate. In the particle swarm optimization algorithm, the upper and lower limits of each particle's position vector were set based on onsite ore dressing technology at 50 and 80, respectively. The final result reveals a time delay of 58 s. Therefore, the dataset can be aligned on a time scale and used as the training and testing sets for the prediction model.

**Table 3  Comparison of predictive model outcomes using different timing delay methods.**

| Method | MAE | RMSE | MAPE | $R^2$ |
|---|---|---|---|---|
| Traditional | 0.04521 | 0.00056 | 0.18972 | 0.78 |
| Improvement | 0.01318 | 0.00004 | 0.21841 | 0.89 |

**Table 4  Evaluation of model detection results.**

| Model | MAE | RMSE | MAPE | $R^2$ |
|---|---|---|---|---|
| $\varepsilon$-SVR | 0.06335 | 0.00012 | 0.04512 | 0.76 |
| GA-MLP | 0.04279 | 0.00013 | 0.03952 | 0.79 |
| $\varepsilon$-GASVR | 0.01318 | 0.00004 | 0.21841 | 0.89 |

To demonstrate the impact of the improved timing delay analysis method on the predictive outcomes, this article separately employs both the traditional timing delay analysis method and the improved timing delay analysis method to perform delay compensation on the same dataset. The results, as presented in Table 3, after applying these methods to the $\varepsilon$-GASVR predictive model, show that the traditional time delay analysis method resulted in a mean absolute error (MAE) of 0.04521, a root mean square error (RMSE) of 0.00056, a mean absolute percentage error (MAPE) of 0.18972, and an $R^2$ value of 0.68. The dataset derived using the particle swarm optimized time delay analysis method provided an MAE of 0.1318, an RMSE of 0.00004, a MAPE of 0.21841, and an $R^2$ of 0.89. This comparison indicates that the model predictive performance using the particle swarm optimized time delay analysis surpasses that of the traditional method, thereby providing a more accurate dataset for the predictive model.

Table 4 reports the evaluation metrics for the high-pressure roller mill particle size qualification rate prediction models established by the three algorithms. Table 3 highlights that the proposed $\varepsilon$-GASVR model has the smallest MAE of 0.01834 and RMSE of 0.00004, the lowest among the three algorithms. Additionally, $R^2$ is 0.89, closer to the ideal value of 1. All the evaluation metrics of the proposed method are superior to the competitor algorithms, indicating that the proposed support vector machine high-pressure roller mill particle size qualification rate detection model has the highest precision, strongest robustness, and most appealing performance. The predictive performance curves of the three prediction models are shown in Figs. 3, 4 and 5, respectively.

# DISCUSSION

## End-edge-cloud system

With the development of industrial IoT and the era of industrial intelligence transformation, cloud computing, cloud storage, and artificial intelligence algorithm technologies are developing rapidly. End-edge-cloud collaboration is a computing architecture that combines edge computing and cloud computing, aiming to achieve collaborative work between edge devices and the cloud (*Zhou et al., 2021*; *Hashem et al., 2015*). This collaborative work typically includes data collection, processing, analysis, storage, and

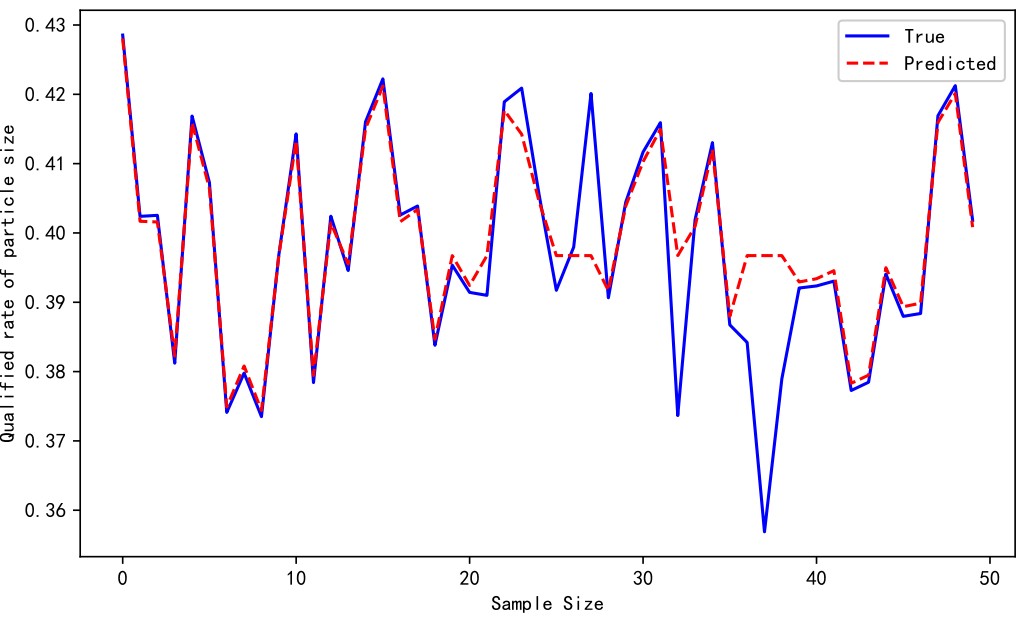

**Figure 3** Prediction performance of GA-MLP.

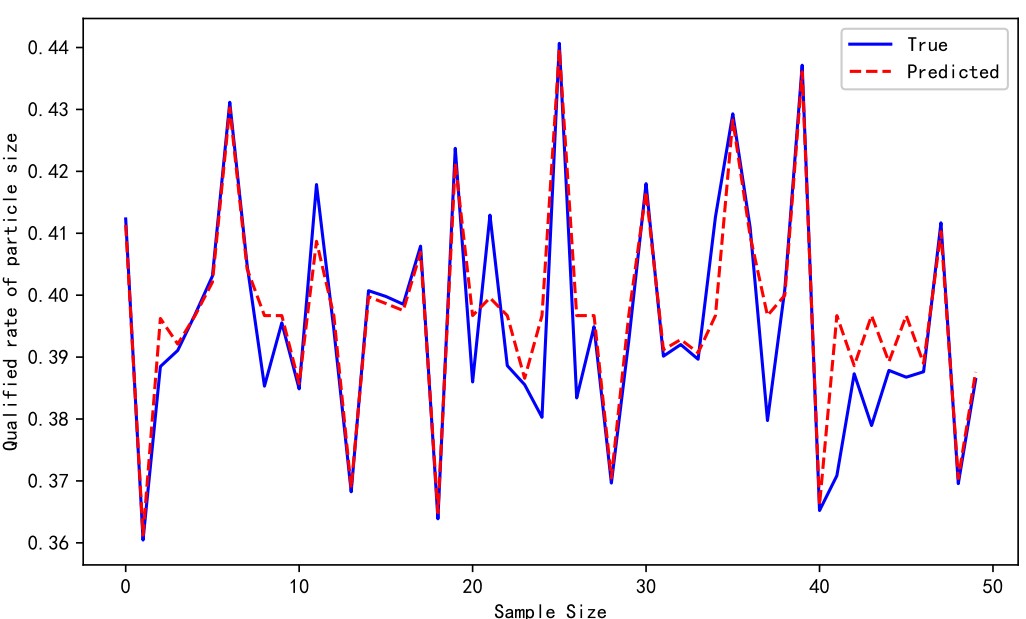

**Figure 4** Prediction performance of $\varepsilon$-SVR.

information sharing between edge devices and the cloud. Thus, this article uses end-edge-cloud collaboration technology and intelligent detection methods to develop a system for monitoring high-pressure roller mills' particle size qualification rate.

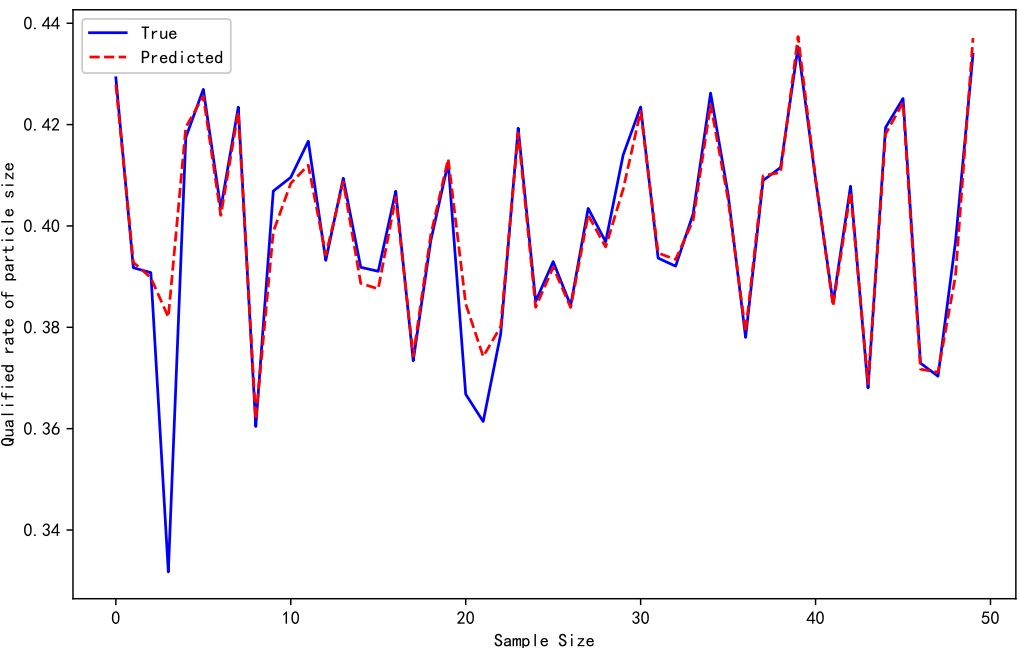

**Figure 5  Prediction performance of $\varepsilon$-GASVR.**

As depicted in Fig. 6, the developed method for testing the qualified rate of particle size of a high-pressure roller mill adopts the end-edge cloud collaborative architecture. The proposed method is a distributed computing architecture that integrates the end, edge, and cloud sides at three levels to achieve efficient collaboration between edge computing and cloud computing. This architecture encompasses the data collection and transmission modules at the end, the initial data processing and the $\varepsilon$-GASVR-based granularity pass rate prediction model at the edge, and the data storage and online self-calibration model in the cloud.

At the end, the core task of the data collection and transmission modules is to gather data in real-time from terminal devices, such as high-pressure roller mills. Additionally, this data is rapidly transmitted to the edge for further processing and then to the cloud for long-term storage. This strategy ensures the timeliness and accuracy of data collection.

At the edge, serving as an intermediary in data processing, it swiftly responds to changes in the status of terminal devices. By linking the terminal and the cloud, the edge module locally performs data storage and preprocessing, and conducts online predictions of granularity pass rate. The model utilizes preprocessed data from the edge layer for predictive analysis and outputs evaluations of the high-pressure roller mill's granularity pass rate. It also uploads processed data and predictive results to the cloud for subsequent self-calibration models and overall performance monitoring.

In the cloud, it focuses on centralized data storage and running the self-calibration model. At this level, an online self-calibration model for granularity pass rate, dependent on the cloud's robust computational resources, is established and maintained. It is responsible for

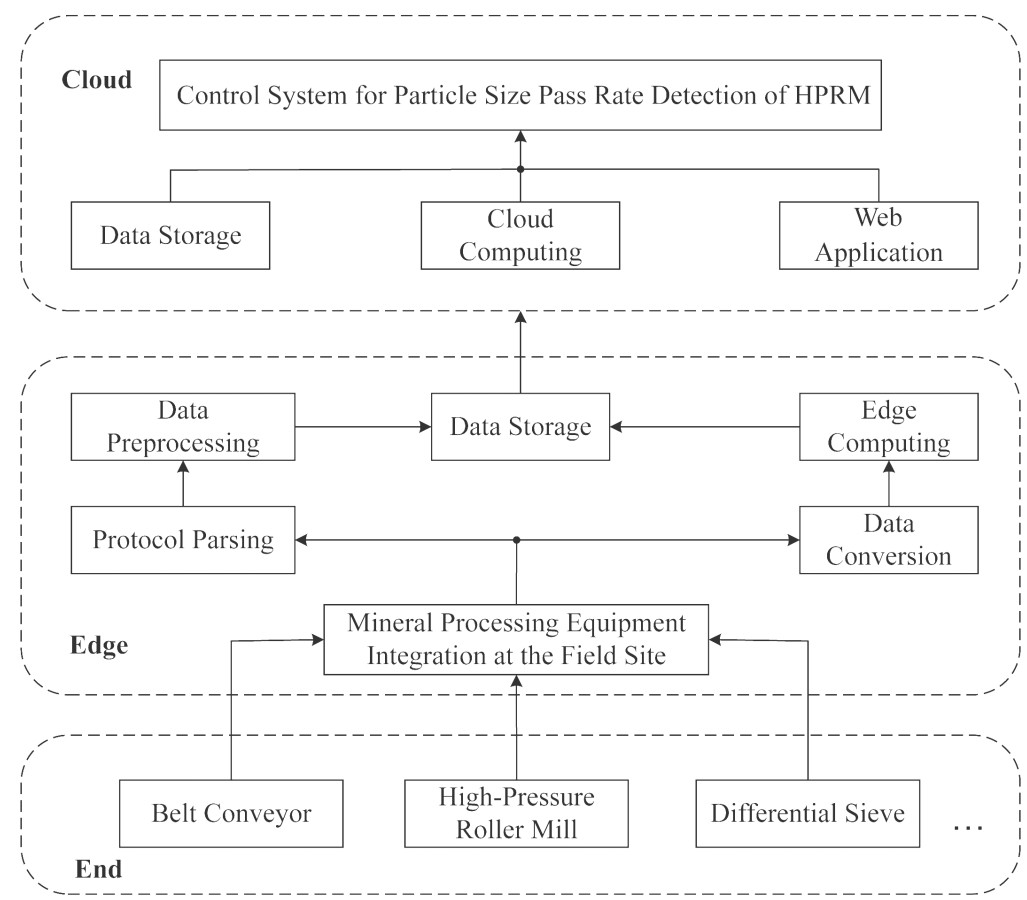

**Figure 6  End edge cloud architecture.**

deep analysis of the predictive model's results and modifications to the model parameters. As the granularity pass rate prediction model needs to adapt to new data over time, it requires regular updates and maintenance while continuously monitoring and evaluating performance to ensure the predictive model's quality and accuracy.

In this end-edge-cloud collaborative architecture, continuous adjustments and optimizations of the granularity pass rate prediction model ensure the system operates efficiently and accurately. By combining the powerful analytical capabilities of cloud computing with the real-time rapid response of edge computing, the end-edge-cloud collaborative architecture fully leverages these advantages to achieve rapid response times and high-quality predictive performance.

## Hardware platform

The intelligent detection system designed to assess the particle size qualification rate in high-pressure roller mills operates on a hardware platform illustrated in Fig. 7. The operational efficiency of the granularity pass rate intelligent detection system for high-pressure roller mills on the hardware platform is crucial. The system utilizes the Siemens

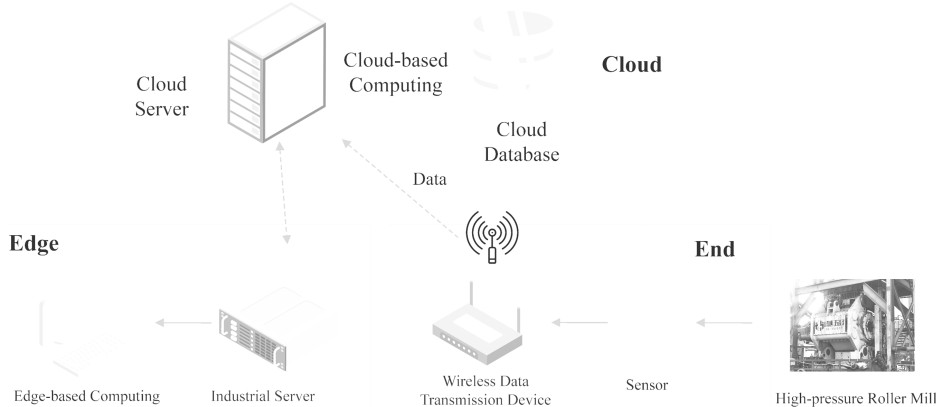

**Figure 7** **End edge cloud system hardware platform.** Source credit: data server icons created by vectorsmarket15 (Flaticon); server icons created by vectorsmarket15 (Flaticon); 3D computer icons created by Freepik (Flaticon); sensor icons created by netscript (Flaticon); sensor icons created by Ylivdesign (Flaticon); database icons created by Vectorslab (Flaticon); wireless router icons created by vectorsmarket15 (Flaticon).

S7-1500 model PLC as the core device for data collection, which is highly reliable and precise, essential for enhancing detection accuracy. This PLC is connected to a variety of sensors on the high-pressure roller mill, enabling real-time monitoring and collection of critical operational parameters, such as pressure and roller frequency. These capabilities facilitate immediate predictions of the granularity pass rate of the high-pressure roller mill. The use of the Siemens S7-1500 not only optimizes the management of data flow but also ensures high efficiency and low latency in data transmission through its advanced processing capabilities, thereby further enhancing the overall system's response speed and reliability.At the edge, edge servers are employed to perform data preprocessing and operate the granularity pass rate prediction model. On the cloud side, cloud servers are utilized for cloud-based data storage and to run the self-calibration model for the granularity pass rate.

## Software platform

The software infrastructure for this system is uniformly operated on the Windows platform. At the end, or sensor-interface level, data acquisition and transfer functions are managed by a module developed with a MySQL database backend and Python scripting for handling the data. For making predictions at the edge, the module utilizes an integrated approach combining Python for backend processing and JavaScript, CSS, and HTML for frontend development, enabling dynamic user interfaces and visualization. In the cloud, an online self-calibration module has been implemented, utilizing the MySQL database for data management, and employing Python scripts to carry out the self-calibration process.

## Application effect analysis

Figure 8 showcases the high-pressure roller mill situated on-site at the designated beneficiation plant in Liaoning, where the predictive technique presented in this study

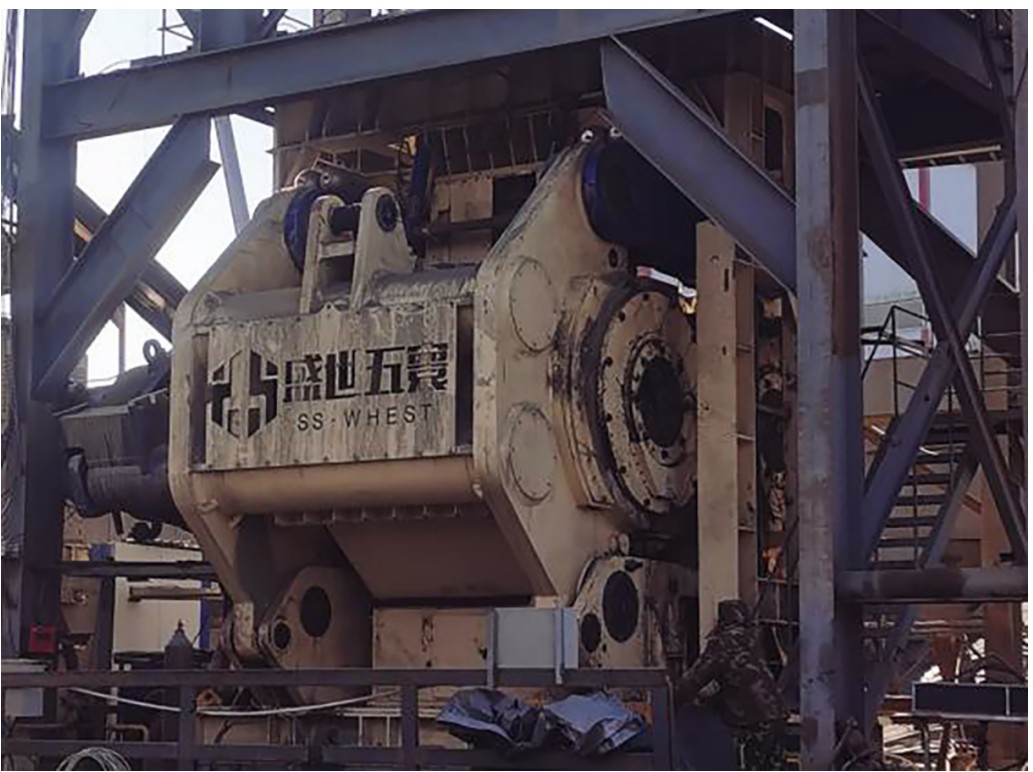

**Figure 8** **High-pressure roller grinder at the ore beneficiation site.**

is put into practice. This approach is utilized specifically for forecasting the particle size qualification rate of the on-site high-pressure roller mill. Given the pivotal role of the high-pressure roller mill in the mineral processing workflow, the ability to anticipate its performance is invaluable for enhancing production efficiency and elevating the caliber of the final product. The application of the intelligent prediction strategy delineated in this article leverages the end-edge-cloud architecture depicted in Fig. 6 alongside the procedural outline provided in Fig. 9.

Figure 10 illustrates the performance of the high-pressure roller mill in terms of the particle size qualification rate, with delay differences between predicted and actual values eliminated. The improved parameters for the online support vector machine prediction model are derived from the experimental modeling data described above. Within the $\varepsilon$-GASVR model, the genetic algorithm parameters include a population size of 500, a crossover probability of 0.8, a mutation probability of 0.05, and the number of iterations set to 1000. The kernel function chosen for the model is the radial basis function kernel, with the tolerance interval set to 0.001. This model receives sensor data at 2-second intervals, which, after undergoing online preprocessing, are fed into the model. As more online data

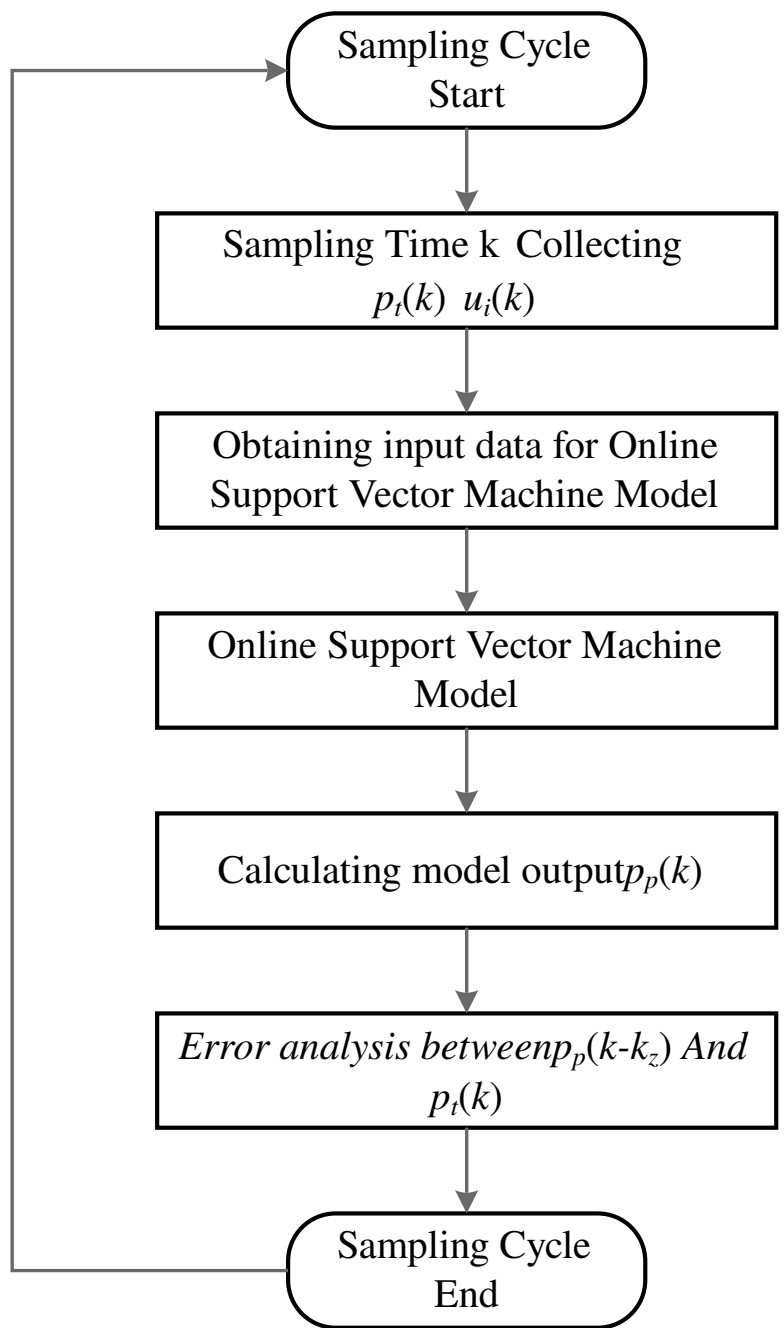

**Figure 9** Online detection process flowchart.

is accumulated, the model undergoes continuous adjustments to enhance its predictive accuracy.

The online model receives data vectors every two seconds, and experimental results indicate that the online prediction accuracy can reach over 95%. As demonstrated in Table 5, the $\varepsilon$-GASVR model, which employs the RMSprop algorithm for hyperparameter

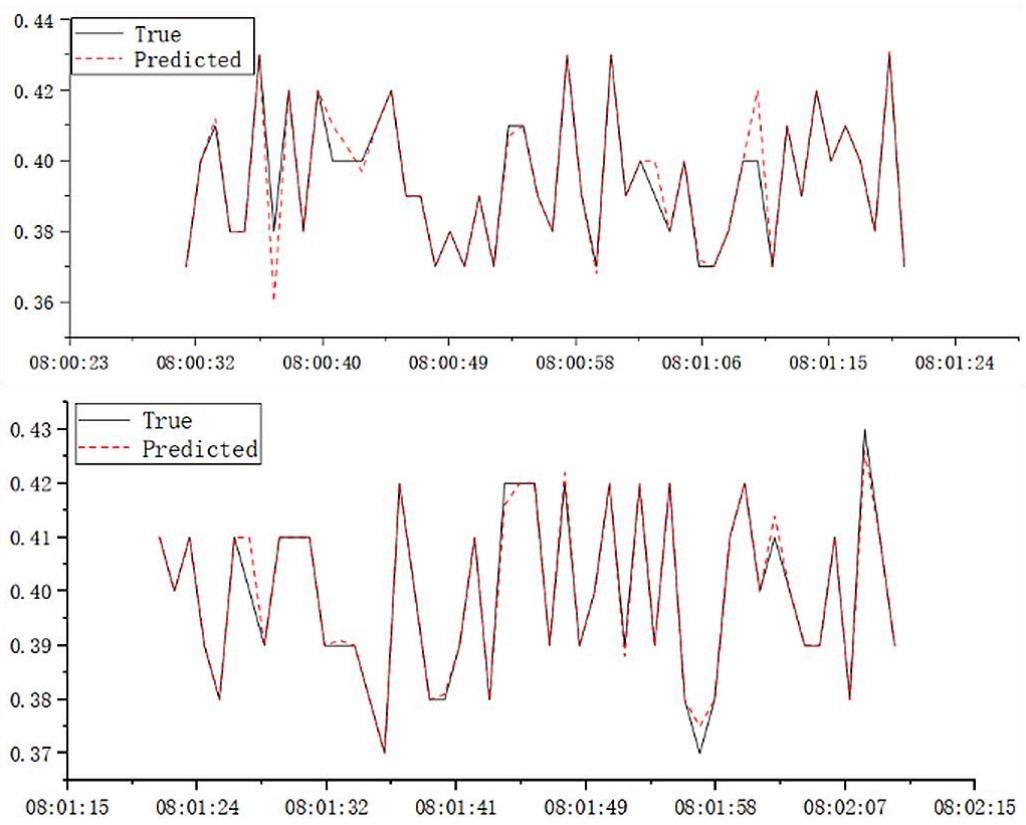

**Figure 10** Online prediction of particle size pass rate for high-pressure roller mill.

**Table 5** Comparison of the online model's accuracy for granularity pass rates.

| Model | TPR | TNR | $R^2$ |
|---|---|---|---|
| $\varepsilon$-GASVR-RMS | 97.24% | 98.53% | 92.62% |
| $\varepsilon$-GASVR | 93.62% | 94.05% | 89.85% |

tuning and incorporates cloud-side self-calibration, achieves a true positive rate (TPR) of 97.24% during the online prediction process. This represents a 3.86% improvement over the $\varepsilon$-GASVR's rate of 93.62%. Moreover, the true negative rate (TNR) of $\varepsilon$-GASVR-RMS reaches 98.53%, which is a 4.76% increase compared to $\varepsilon$-GASVR's 94.05%. The $R^2$ score for the $\varepsilon$-GASVR model, also using the RMSprop for hyperparameter adjustments, achieves 92.62% in online prediction settings, marking a 3.08% enhancement from the $\varepsilon$-GASVR's score of 89.85%. These outcomes substantiate the superior real-time performance and accuracy of the $\varepsilon$-GASVR online prediction model for assessing granularity pass rates in high-pressure roller mill systems.

The end-edge-cloud architecture significantly reduces latency caused by long-distance data transmission compared to traditional end-to-cloud architectures by dispersing data processing tasks across edge devices. This structure enables edge devices to process data instantaneously, thereby substantially enhancing response times and strengthening the

**Table 6 Comparison of single data transmission response time between end cloud system and end edge cloud system.**

| Response time (ms) | Sample 1 | Sample 2 | Sample 3 | Sample 4 | Sample 5 | Sample 6 |
|---|---|---|---|---|---|---|
| End-Edge-Cloud | 173 | 167 | 166 | 170 | 164 | 165 |
| End-Cloud | 716 | 752 | 759 | 742 | 717 | 771 |

real-time processing capabilities of online prediction models. Additionally, the end-edge-cloud architecture enhances data security and privacy protection during transmission and processing. Data is primarily processed locally, with only essential information being sent to the cloud, thus minimizing the risk of data exposure and providing stronger data protection and privacy security. Table 6 presents a comparison of response times for single data transmissions and prediction outcomes under end-edge-cloud and end-cloud architectures.

Table 6 presents a comparative analysis of the response times at six distinct sampling points for both the end-edge-cloud system and the end-cloud system. The end-edge-cloud architecture achieves faster response times by decentralizing data processing tasks to edge devices, thereby reducing the total distance and time required for data transmission. In contrast, the end-cloud architecture involves data transmission over longer distances to cloud servers, which increases the overall time for data transfer and processing. The data from Table 6 indicates that response times for the end-edge-cloud system range from 164 ms to 173 ms. The response times for the end-cloud system are significantly longer, varying from 716 ms to 771 ms. The end-edge-cloud system exhibits a marked advantage in response time over the end-cloud system, reducing latency by approximately 78%. This advantage makes the end-edge-cloud architecture more suitable for scenarios that require rapid response and real-time processing. By leveraging edge computing, there is a significant enhancement in the overall efficiency and performance of the system, particularly in applications like the prediction of acceptable particle size in high-pressure grinding rolls, where the timeliness and accuracy of data processing are crucial.

During the software development phase, the backend establishes a connection to the database *via* the MySQL package, where the specifically designed data tables are stored. This backend retrieves the necessary data from these tables, converting the data into JSON format for further processing. The data in JSON format is then transmitted to the frontend using the Python Flask framework's API endpoints. In turn, the frontend utilizes the appropriate APIs to fetch the data provided by the backend, displaying it within the relevant sections of the custom-designed frontend interface. This monitoring interface on the frontend is crafted with HTML, CSS, and JavaScript, enhanced by the use of the jQuery component library and Echarts for graphical visualizations.

Ultimately, the prediction model is implemented on an Industrial Internet platform. The outcomes of the model predictions and the software's overall performance are depicted in Fig. 11. This figure designates the current time on the $x$-axis and the particle size qualification rate on the $y$-axis while portraying the temporal progression of the particle size qualification rate through a curve. A comparison between the predicted and actual

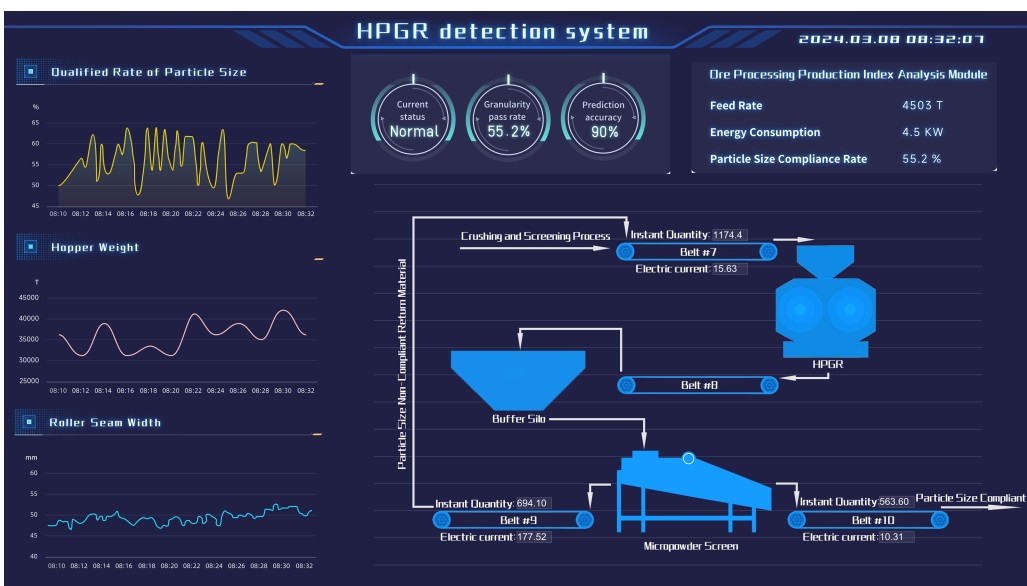

**Figure 11** The software interface of the high-pressure roller mill particle size pass rate prediction system.

particle size qualification rates allows for an evaluation of the operational stability of the high-pressure roller mill. Such assessments are crucial in averting industrial accidents arising from issues with equipment reliability.

The particle size qualification rate detection software interface is depicted in the accompanying figure. The software is designed to gather and store critical on-site operational data—such as the weight of the material warehouse, the width of the roll gap, clamping pressure, and roll frequency—directly from the beneficiation plant into the database. Once the data is collected, it undergoes a preprocessing step to ensure quality and accuracy. Following preprocessing, the software calibrates the particle size qualification rate prediction model on both the edge and cloud levels. Subsequently, it predicts the current particle size qualification rate using this model. The predictive results, including calculations of the rate, are then saved back into the database. These outcomes are visualized in the software interface, where they are presented both graphically, in chart form, and numerically, to enable easy interpretation by the users.

# CONCLUSIONS

This study developed a predictive model for a high-pressure roller mill's particle size qualification rate, which is optimized based on the end-edge-cloud collaborative architecture. In the experimental process, the processing sensor data of the high-pressure roller mill was analyzed in detail, and the particle swarm optimization time delay analysis algorithm was used to eliminate the time difference between the particle size qualification rate and other data. The time delay result obtained through the experiment is 58 s, and the data was aligned on this time scale to eliminate the time difference between the particle size

qualification rate and other data. Based on the aligned dataset, the maximum information coefficient (MIC) was used to analyze the correlation between each variable and the qualified rate of granularity, and key features were selected for model training. Finally, the feature inputs were selected: the seam width, bin weight, roll frequency, locking pressure, and processing capacity. The developed method used an improved genetic algorithm to establish an $\varepsilon$-GASVR offline prediction model, then migrated the model to the online system and combined it with the RMSprop optimization algorithm for real-time model updates. After analyzing the integrated time delay, the resulting delay time was 58 s.

The proposed predictive system can evaluate the processing status of the high-pressure roller mill, adjusting its operating parameters to ensure that the particle size reaches the qualified standard. The experiment demonstrates that $R^2$ of the predictive model achieved 0.89, effectively providing a reliable basis for closed-loop optimization control of the high-pressure roller mill. The presented method can support energy-saving and efficient production and demonstrates the potential of using advanced algorithms in industrial applications.

### Funding
The authors received no funding for this work.

### Competing Interests
The authors declare there are no competing interests.

### Author Contributions
- Hairong Guo conceived and designed the experiments, performed the experiments, analyzed the data, performed the computation work, prepared figures and/or tables, authored or reviewed drafts of the article, and approved the final draft.
- MingYin Yan conceived and designed the experiments, performed the computation work, authored or reviewed drafts of the article, and approved the final draft.
- Jing Zhao conceived and designed the experiments, analyzed the data, authored or reviewed drafts of the article, and approved the final draft.
- Lanhao Wang conceived and designed the experiments, performed the experiments, authored or reviewed drafts of the article, and approved the final draft.

### Data Availability
The raw measurements are available in the Supplementary Files.

### Supplemental Information
Supplemental information for this article can be found online at http://dx.doi.org/10.7717/peerj-cs.2151#supplemental-information.

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
