# Peer review of "Prediction method of particle size qualification rate of high-pressure roller mill based on end-edge-cloud synergy"

_PeerJ Computer Science, doi:10.7717/peerj-cs.2151_

## Round 0.1 · original submission · Major Revisions

- The research problem should be highlighted and discussed well in the introduction, supported by up-to-date references.

- The research contributions should be presented well.

- In the results, review and discuss the time delay analysis findings, and align it with the research probelm and objectives.

- Proofread the paper.

**Language Note:** The Academic Editor has identified that the English language must be improved. PeerJ can provide language editing services - please contact us at [email protected] for pricing (be sure to provide your manuscript number and title). Alternatively, you should make your own arrangements to improve the language quality and provide details in your response letter. – PeerJ Staff

Reviewer 1 ·

Basic reporting

This article addresses the crucial aspect of pass rate detection in high-pressure grinding processes. Traditionally reliant on manual judgments or offline inspections, the proposed method introduces a time-delay analysis leveraging the particle swarm optimization algorithm to mitigate delays in pass rate data compared to other variables. The study incorporates Maximum Information Coefficient correlation analysis to select pertinent features, optimizing an offline support vector machine model through genetic algorithms. An online prediction model is then established, utilizing Root Mean Square Propagation gradient descent optimization for real-time pass rate predictions and model updates. This intelligent detection mechanism enhances granule pass rate monitoring in high-pressure grinding machines, integrating end-edge-cloud collaboration for comprehensive efficiency.
However, the following comments need to be taken into the considerations:
1. The researcher need to improve the literature review.
2. Explain the functions of the equations in the relation to the title of the manuscript.
3. Please include the major result in the abstract.

Experimental design

As above

Validity of the findings

As above

Reviewer 2 ·

Basic reporting

English used is not too bad, it is readable however, there are many typo mistakes, for example, all the in-text citations required on space before the brackets of the reference (e.g., line 36, 46, 72,75,..etc).

In terms of Literature review, no literature review includes all the papers that have been done before and highlights the research gap that needs to be addressed in this work. The introduction and background section, have a poor in-text citation and references. Many references need to be added to that section as most of the content there is general information not part of the contribution.
The article structure is missing the Literature review as mentioned before. I suggest using section 2 for the Literature review. Figures, most of them need to be enhanced with high quality mainly figures number 2,5, 10,11, and 12.

Results; all the metrics used for evaluation are explained well. The predicted models have been presented and explained. However the authors said that this paper aims to propose time delay analysis to eliminate the delay impact between the pass rate of granules and other data; there are no results presented to discuss this point. Moreover, I was excited to know how the prediction models were applied in a cloud and edge, but unfortunately in the discussion section, I found general information with citations about cloud and edge that was previously mentioned in section 2, and it did not add anything.

Experimental design

The research objective is proportional to the aim of the journal. The research topic is important and interesting, it has a well-defined goal: designing a framework that uses cloud and edge to detect and control the particle size qualification rate of high-pressure grinding rolls. However, the research gap addressed in this work is not clear because there is no research gap defined based on previous work.

Method
There is a clear description of the proposed architecture for predicting the qualified rate of particle size in high-pressure roller mills based on edge-cloud collaboration. Most of the stages required for this architecture are explained in details. However, the part of AI model implementation was not explained well. In more precise, how SVM used in this work was implemented, what are the parameters used with this algorithm, size of the datasets. Also, the architecture mentioned the sensors for data collection, does the data collected in real-time using these sensors or historical data is used (offline). Moreover, the communication between the three layers of the proposed architecture is not clearly explained or mentioned.
On the other hand, a Genetic algorithm has been used to optimize the parameters, but it would be much better to summarize the GA parameters such as Population size, Generation, crossover population, and mutation probability.

Validity of the findings

The paper contains important work that can be published, but it needs to be improved and modified according to the comments described previously.

In general, I found that the authors focused on the theoretical aspect more broadly, as an extensive time analysis including a large set of mathematical equations which is not part of the contribution. Instead, if the proposed architecture were better explained and discussed, it would be more suitable for publication. Moreover, a validation model needs to be applied for the results or a comparison with other works needs to be done.

The Conclusion needs to be rewritten better so that the contributions are clearly explained, and the results obtained must be highlighted and provided with numbers, or percentages.

Annotated reviews are not available for download in order to protect the identity of reviewers who chose to remain anonymous.

---

## Round 0.2 · Minor Revisions

I thank the authors for their efforts to improve the work. However, the reviewer is not satisfied with it. Some issues are not issued. For this round, please carefully revise the article and make sure to address the issues thoroughly as per the attached PDF comments from R2.

Reviewer 2 ·

Basic reporting

look to my previous report.

Experimental design

look to my previous report.

Validity of the findings

look to my previous report.

Additional comments

look to my previous report.

Annotated reviews are not available for download in order to protect the identity of reviewers who chose to remain anonymous.

---

## Round 0.3 · accepted · Accept

Congratulations to the authors. Your revision satisfied the reviewer successfully. The current version can be accepted.

Reviewer 2 ·

Basic reporting

I can see all the comments raised in the second round have been addressed. The paper has been enhanced and has been more qualified for publication.

Experimental design

No comments

Validity of the findings

No comments